# An Analysis of Blockchain-Based IoT Sensor Network Distributed Denial of Service Attacks

**DOI:** 10.3390/s24103083

**Published:** 2024-05-12

**Authors:** Kithmini Godewatte Arachchige, Philip Branch, Jason But

**Affiliations:** Department of Telecommunications, Electrical, Robotics and Biomedical Engineering, Swinburne University, Melbourne 3122, Australia; kgodewattearachchige@swin.edu.au (K.G.A.); jbut@swin.edu.au (J.B.)

**Keywords:** blockchain, Internet of Things, vulnerabilities, anomalies, security, low-power, wireless, sensors

## Abstract

The Internet of Things (IoT) and blockchain are emerging technologies that have attracted attention in many industries, including healthcare, automotive, and supply chain. IoT networks and devices are typically low-powered and susceptible to cyber intrusions. However, blockchains hold considerable potential for securing low-power IoT networks. Blockchain networks provide security features such as encryption, decentralisation, time stamps, and ledger functions. The integration of blockchain and IoT technologies may address many of the security concerns. However, integrating blockchain with IoT raises several issues, including the security vulnerabilities and anomalies of blockchain-based IoT networks. In this paper, we report on our experiments using our blockchain test bed to demonstrate that blockchains on IoT platforms are vulnerable to DDoS attacks, which can also potentially lead to device hardware failures. We show that a number of anomalies are visible during either a DDoS attack or IoT device failure. In particular, the temperature of IoT hardware devices can exceed 90 °C during a DDoS attack, which could lead to hardware failure and potential fire hazards. We also found that the Block Transaction Rate (BTR) and network block loss percentage can increase due to corrupted hardware, with the BTR dropping to nearly zero blocks/sec and a block loss percentage of over 50 percent for all evaluated blockchains, and as high as 81.3 percent in one case. Our experiments demonstrate that anomalous temperature, latency, bandwidth, BTR, and network block loss percentage can potentially be used to identify DDoS attacks.

## 1. Introduction

The Internet of Things (IoT) has the potential to revolutionise the way products, places, and people connect, enabling industries to use low-power sensors and embedded physical devices such as microcontrollers and single-board computers [1]. Similarly, blockchain technology has significantly evolved over the past decade, providing a robust security solution for many potential applications [1]. With the latest advances in IoT and blockchain technologies, blockchain has the potential to offer security capabilities to protect IoT end devices [2]. The fusion of these two technologies has been referred to as the Blockchain of Things (BCoT). The BCoT has great potential. However, the integration of blockchain and IoT poses several concerns—notably, the identification of blockchain network security vulnerabilities and anomalies [2].

In this paper, we use our test network to assess blockchain network vulnerabilities and behaviours of different blockchain sensor networks when experiencing a DDoS attack [3]. Our experiments reveal that blockchain networks are potentially vulnerable to Distributed Denial of Service (DDoS) attacks, which can lead to network latency and bandwidth usage anomalies [3]. At the same time, our experiments indicate that DDoS attacks can increase hardware temperatures that can lead to hardware failures due to overheating. We also observe that Block Transaction Rate (BTR) and block loss anomalies also can be visible as a consequence of hardware failure [3].

The contribution of this research is to help identify potential security vulnerabilities that may cause cyber security threats on blockchain-based IoT systems [4]. Additionally, the anomalies visible during cyber security threats can be used to recognise potential future cyber intrusions [5].

The issue of data integrity and user privacy in aged care has emerged as a matter of grave concern in recent years. This has prompted the Australian Government to conduct a Royal Commission into the sector [6]. The report presented by the commission is disconcerting as it uncovers instances of neglect and abuse of elderly individuals in aged care facilities. The findings of the commission highlight the need for a comprehensive overhaul of the aged care sector in the country [6]. As such, it is imperative for the government to prioritise the development and implementation of measures that ensure the safety and well-being of the elderly in aged care facilities.

One of the key recommendations is to better understand the user benefits of blockchain technology and security challenges of commercially available blockchains: “Blockchain technology, however, must also be understood as the foundation of a much broader technology stack including artificial intelligence and the internet of things. As such, while our submission focuses on the regulatory tensions presented by blockchain, it is important to recognise that ameliorating these challenges powers-up a much broader tech stack, contributing to the development of Australia” [7].

Senator Andrew Bragg, Chairman of the Select Committee on Australia as a Technology and Financial Centre, also recommended to the Australian government the following: “Traditional sectors, such as agriculture and education, are integrating digital technologies into their operations in efforts to improve productivity. More deeply, entrepreneurs are developing new business models that are natively digital, built using technologies such as blockchains, smart contracts and machine learning. This technology stack presents an unprecedented opportunity to build a modern digital Australian economy” in senate submission 67 [7]. Also, the Australian government released $3 million in funds under “Blockchain Pilot Grants” and “Australia’s Blockchain Roadmap” programs to support industry transformation [8].

Considering the importance of potential security vulnerabilities and anomalies of blockchain-based IoT sensor networks, to bridge the existing research gap we developed a real blockchain-based test bed using three commonly used blockchain platforms and our network comprises multiple IoT devices [9].

The remainder of this paper is structured as follows. We discuss blockchain technology in Section 2, and we discuss blockchain technology and IoT in Section 3. Section 4 outlines the related work. Section 5 illustrates research methodology and test bed development. We present the results and evaluation in Section 6. Section 7 concludes the paper and illustrates future research work.

## 2. Blockchain

Blockchains are Peer-to-Peer (P2P) networks that make use of Distributed Ledger Technology (DLT) to record transactions across a network [10]. Blockchains are typically transparent, immutable, and secure. Blockchains generate a chain of blocks with each block containing a number of transactions [10]. Once a block is added to the chain, the block cannot be altered. Blockchains are characterised by decentralisation, hash functions, cryptography, smart contracts, and time stamps. Blockchains can be categorised into five main network types based on their services [11].

### 2.1. Public Blockchain Networks

Public blockchains are permissionless blockchains that provide users unrestricted read and write functions and network access. Public blockchains are self-governed networks where any user can communicate over the network [11]. Public blockchains have global accessibility, transparent block transactions, and comparatively higher interoperability. Also, many Public blockchains use tokenisation to identify the ownership of the blockchain account as a security precaution. Public blockchains are typically community-driven networks, and community developers often contribute to their developments [11].

### 2.2. Private Blockchain Networks

Private blockchains are permissioned blockchains that provide restricted network access to blockchain users [11]. These blockchain networks are mostly used within organisations where only a limited number of blockchain users are participants. Private blockchains only allow certain authorised users [11]. Private blockchains are centrally governed networks that comply with organisational regulations. Also, Private blockchains are highly customisable and integrate with existing organisational systems [11].

### 2.3. Hybrid Blockchain Networks

Hybrid blockchains have both public and private blockchain features. Public blockchains allow network access to any user [11]. However, only certain blockchain users have access to certain blockchain services. Hybrid blockchains can be centralised or decentralised depending on the organisation’s requirements. Also, Hybrid blockchains are customisable and provide anonymity [11]. Hybrid blockchains are comparatively low-cost and have more flexibility in security policies. These blockchains are more suitable for projects or organisations that require higher public trust, such as the banking industry [11].

### 2.4. Consortium Blockchain Networks

Consortium blockchains consist of both public and private blockchain components, where multiple organisations manage a single blockchain network [12]. Unlike Public blockchains, which allow anyone to participate in the process of transaction verification and block addition, consortium blockchains restrict this participation to a pre-selected set of nodes [12]. This approach combines the transparency and security benefits of blockchain technology with the control and efficiency desired by enterprises. Moreover, Consortium blockchains are ideal for facilitating collaboration among various organisations [12].

### 2.5. Sidechain Networks

Sidechains are a recently developed type of blockchain where additional blockchains run parallel to the main blockchain, allowing for more functionality and scalability [12]. A sidechain is a separate blockchain that is attached to a mainchain. Sidechains can operate under different rules and consensus mechanisms than the mainchain, providing flexibility and customisation for specific use cases [12]. They are particularly valuable for offloading transactions from congested networks, testing updates or new features in a sandbox environment, and enabling communication and asset exchange between disparate blockchain systems [12].

## 3. Blockchain Technology and IoT

IoT devices and networks connect sensors and actuators to collect and process information, supporting the community in living and working smarter [13]. The deployment of IoT contributes to a wide range of industrial developments, such as smart vehicles, IoT wearable devices, IoT agriculture tools and supply chain tracking devices. Ultimately, IoT usage leads to the automation of processes and reduces costs. It also improves the quality of services [14].

However, as IoT uses low-power devices, they are susceptible to cyber intrusions [15]. Researchers have identified blockchain as a possible solution to secure IoT devices and networks. This has led to the development of the “Blockchain of Things” by integrating blockchain and IoT technologies [16].

The primary intention of the Blockchain of Things (BCoT) is to enhance the security and performance of the IoT sector [14]. To accomplish this, BCoT has introduced four key parameters. They can be stated as follows.

### 3.1. Interoperability

Interoperability between IoT smart devices and low-power microcontroller devices is crucial for the smooth operation of cyber physical systems. This is particularly important for transmitting data over blockchain IoT networks [14]. To achieve this interoperability, a blockchain-composite layer is commonly used among blockchain-based IoT nodes. Additionally, the BCoT ensures consistent access to peer-to-peer sensor networks through blockchain interoperability [14,15].

### 3.2. Traceability

The ability to track and verify data in an IoT blockchain is crucial for improving the performance of BCoT [14]. It is vital to have a system in place that allows for the tracing and verification of blockchain data in a network that is based on both blockchain and IoT. Additionally, blockchain technology provides timestamps for each IoT node, ensuring that the nodes have been traced and verified [14].

### 3.3. Reliability

The reliability of IoT data refers to the quality and trustworthiness of the information [14]. Blockchain technology is used to ensure the security and integrity of this data by employing encryption algorithms, digital signatures, and timestamps [16]. By enhancing the reliability of blockchain data, the performance of data transmission over blockchain networks is improved [14,15].

### 3.4. Autonomic Interactions

The use of autonomic interactions improves the capabilities of IoT systems by connecting them with reliable blockchain networks while avoiding untrusted third-party networks [14]. BCoT utilises smart contracts to enable these autonomic interactions, enhancing blockchain performance on low-power IoT devices. These interactions occur between sensor nodes within the blockchain [14].

## 4. Related Work

According to Hong Ning Dai et al., IoT automated systems can be susceptible to cyber intrusions due to poor interoperability, low processing power, and unsecured data transmissions [14]. The authors suggest that blockchain technology can be used in IoT systems and have proposed a new architecture called Blockchain of Things (BCoT) [14]. Furthermore, the authors have analysed the use of blockchain technology with 5G cellular connections. As the authors have highlighted, the use of IoT cyber physical systems can be challenging due to the diversity of IoT devices and systems, network complexity, heterogeneity of IoT data, and limited resources [14]. The paper emphasises that blockchain technology can help overcome these challenges. As per the authors, blockchain technology can enable the validation of IoT data and use a blockchain-based mutually distrusted cyber system to validate data using BCoT [14].

Weidong Fang et al. proposed a blockchain trust model (BTM) to identify malicious nodes in a blockchain IoT network [17]. Identifying malicious nodes in a blockchain IoT network is vital for ensuring the network’s security. The proposed model uses the blockchain concept to construct a data structure and is tested using low-power IoT devices in real-time [17]. They have used a 3D space to record network performance, demonstrating that the developed model can effectively detect malicious blockchain nodes. To identify and eliminate external and internal attacks, they used an IoT smart architecture involving Wireless Sensor Networks (WSN) and the Neighbour Weight Trust algorithm (NWTD) [17]. The NWTD gathers information about the minimum acceptable threshold of the trust nodes and separates infected blockchain nodes from the core network. The blockchain platform ensures the validity of data packets and periodically assists the trust threshold [17].

The NWTD platform also provides a clear distinction between malicious and trustworthy nodes by using a deidentification and elimination method [17]. Authentication and trust management systems are driven through a secure platform to ensure data confidentiality and validity. The system uses node authentication and has the ability to find the trustor and trustee using IoT end devices [17]. Blockchain applications focus on the privacy of the data system and managing user information and control in the context of the blockchain platform. It has the capability of controlling blocks without overloading the IoT network and provides storage for trust information [17].

Michail Sidorov et al. proposed a system that combines blockchain and IoT technologies for autonomous monitoring and structure control simulation [18]. The system is designed to enable independent decision-making, secure information sharing, and transparency. It leverages Structural Health Monitoring (SHM) and distributes it across edge and core networks [18]. The IoT platform and blockchain application are connected to sensors and physically attached to the SHM, ensuring safe operation. The sensors collect data and transmit it through a gateway to the main servers. The authors highlight that IoT end devices are capable of providing real-time information with high accuracy and responding to emergencies [18]. The system includes a damage detection mechanism that inspects routine operations by eliminating critical signs of damage [18]. This process protects information integrity and enables crucial decision-making. The paper addresses previous system issues, such as bottleneck bandwidth and single points of failure, by eliminating central network architectures and replacing them with distributed or decentralised network architectures, which are less vulnerable to cyber-attacks [18].

According to Kevin Jonathan et al., Bitcoin is a commonly used blockchain platform in various financial markets and services [19]. More than 3000 financial organisations use Bitcoin as their main financial transaction platform. The authors have highlighted that although blockchain has its own security measures, 51% of cybercrimes reported in 2019 involved blockchain technologies [19]. Their paper emphasises the potential security issues and vulnerabilities related to blockchain technology. The authors compared energy consumption, scalability, network performance, consensus confirmation time, and block creation time for different blockchain platforms using Re-entrancy attacks and Majority attacks [19]. They simulated the attacks and exploited the Bitcoin and Ethereum blockchain networks. As per the authors, the Majority attack can severely damage Bitcoin mining pools and hardware computing power, while the Re-entrancy attack involves exploiting smart contracts [19]. The paper highlights that potential damages to blockchain financial transactions can raise economic concerns [19].

According to Houshyar Honar Pajooh et al., the Hyperledger Fabric blockchain is a resource-constrained platform that is capable of safeguarding low-power IoT devices while detecting network anomalies [20]. The Hyperledger Fabric blockchain applications not only provide a high level of security, privacy, and data integrity but also offer robustness to network users [20]. The transparency of data transactions in blockchain applications makes it easier to identify network anomalies. A centralised peer-to-peer network architecture is commonly used in IoT networks, which is vulnerable to single-point failures [20]. Nevertheless, this can be avoided by adopting a decentralised network architecture, which is a fundamental security measure for a blockchain network. Decentralised network architectures can also prevent availability attacks such as denial-of-service attacks [20].

Tomasz Hyla et al. note that with the increasing prevalence of cyberattacks and cybercrimes, it has become imperative to ensure that digital health systems are well secured and have the ability to detect abnormal behaviours [21]. They suggest that permissioned blockchain architecture is a reliable and accountable solution to protect electronic health systems. Modern blockchain applications are designed to identify anomalous behaviours in blockchain networks, making blockchain technology a potential solution to enhance data integrity and accountability [21]. This can be achieved through the implementation of the integrity protection service model, which was developed to ensure blockchain transaction transparency in a permissioned blockchain network [21]. By using this model in IoT networks, it is possible to analyse the security level and performance while detecting anomalous behaviours in IoT networks [21]. As the authors have emphasised, the healthcare sector is especially concerned about data integrity in digital health databases, and a security breach can put lives at risk. Newly developed permissioned blockchain applications use off-chain information storage to minimise fault tolerance [21].

Marko Hölbl et al. proposed that blockchain technology can offer decentralised and distributed network features for IoT networks [22]. This implies that there may not be a necessity for a central authority to verify user access and data transmission. All data transactions are secured through encryption algorithms [22]. According to the authors, blockchain applications have been used in healthcare to secure health data. Blockchain technology is used in health sensor networks to maintain the integrity and authenticity of electronic health records [22]. Their research aims to explore reliable and sustainable blockchain applications to address various cyber challenges in healthcare sensor network environments. The paper also highlights that blockchain technology has opened up new research opportunities, such as evaluating blockchain power consumption and hardware resource utilisation [22]. Modern blockchain applications have integrated biometric authentication to authenticate users and prevent unauthorised access. Additionally, blockchain technology uses a secure block architecture to transmit encrypted data [22]. Although blockchain technology provides additional security features, the power consumption of blockchain platforms may impact hardware performance utilisation. Therefore, it is essential to understand the security capabilities and performance capabilities of blockchain-related technologies [22].

According to Faisal Jamil et al., the advent of blockchain technology has revolutionised the healthcare industry by providing a robust security solution for digital health management platforms [23]. Faisal Jamil et al. note that the use of blockchain technology in healthcare has led to the decentralisation of digital health management platforms, which has significantly enhanced the security of healthcare information [23]. Portable digital devices are now widely utilised by healthcare facilities to improve the quality of medical treatments and health monitoring [23]. Blockchain technology can provide security features for portable digital devices to protect medical data in a connected wireless network. The decentralised network architecture of blockchain plays a fundamental role in safeguarding accountability for sensitive information and the privacy of blockchain users [23].

Elli Androulaki et al. proposed a blockchain-based operating system that utilises a Hyperledger fabric system [24]. The system is designed to monitor the integrity level of network data transmissions and evaluate the secure accessibility of the blockchain network. Health management platforms in healthcare facilities require high-speed networks to avoid data transmission latency [24]. The proposed system utilises data transaction speed, which is known as “Hyperledger Caliper”. This benchmark process empowers resource utilisation mechanisms [24]. The proposed operating system provides a robust security solution for healthcare management platforms, enhancing the privacy and reliability of sensitive medical data [24].

According to Gioele Bigini et al., modern mobile devices have the potential to improve the processes and well-being of the general population [25]. However, due to the increasing number of cyber threats, smart IoT applications and devices are vulnerable to compromise [25]. The authors highlight the effectiveness of blockchain technology in addressing cybersecurity issues in the IoT sector. The authors also emphasise the need for industries to comply with cybersecurity regulations to protect sensitive information. The paper highlights that blockchain technology, with its distributed ledger systems and decentralised network features, can provide a robust security solution to protect information [25]. Decentralisation is particularly useful for low-security networks, as it enhances information integrity and accountability. Additionally, blockchain technology enables users to have full ownership of their information and addresses privacy concerns [25].

Joseph Merhej et al. proposed a blockchain-based technique to exchange health information in a secure and reliable way [26]. The authors emphasise that the Health Information Exchange (HIE) allows medical professionals to share patient medical information and records. However, HIE systems face critical security, latency, scalability, and privacy concerns [26]. The authors proposed an efficient model called “Efficient Healthcare Information Exchange” (ELSO). The ELSO architecture combines public blockchains and an off-chain database to store sensitive information [26]. Public blockchains are used to store patient personal information, while the off-chain database is used to store medical information. The ELSO architecture was designed based on HIPAA and GDPR health information security regulations and comprises three technical layers: the application layer, the control layer, and the storage layer. The application layer controls the patient biometric information, the control layer controls smart contracts and permissions, and the storage layer controls the storage log files and off-chain database [26].

In another work by Joseph Merhej et al., the authors emphasise that the Healthcare Information Exchange (HIE) plays a key role in healthcare and that detecting modified transactions is important [27]. The paper proposed a secure framework called “DeepChain” that combines deep learning and blockchain technology [27]. DeepChain uses two types of blockchains to enhance data security and a deep learning model called the “Generative Adversarial Network” (GAN) to detect risky transactions [27]. The authors tested the performance of the framework based on real health data. The paper addresses the uncovered privacy, performance, and security aspects of HIE systems. DeepChain proposes a four-level data security architecture, which contains the application layer, control layer, detection layer, and storage layer [27]. The GAN model consists of two neural networks called generator and discriminator that are used to distinguish real data from fake data. The GAN generator also uses the MIMIC database to train the deep learning algorithm. The test setup was developed using virtual machines and an HTTP server. The authors used Visual C++ software (Version 12.0.40664.0) to create the graphical interface and Solidity programming language to build blockchain smart contracts [27].

Joseph Merhej et al. also noted that the World Health Organisation has invested a large amount of money to improve data exchange in different healthcare facilities [28]. In particular, this paper conducted a comparative analysis of recent Healthcare Information Exchange (HIE) approaches that consider data security, integrity, privacy, scalability, accuracy, latency, transaction confirmation time and efficiency metrics, which are called the eight pillars [28]. The eight-pillars-based approach is used to identify the best blockchain solution for HIE systems. As the authors have highlighted, HIE systems are vulnerable to numerous threats, including eavesdropping, industrial espionage, lack of physical hardening, and user ignorance due to the heterogeneity of IoT devices [28]. According to the authors, evaluation can also be used efficiently in monitoring elderly homes using a Pyroelectric Infrared sensor (PIR) [28].

At present, only a handful of papers have been published that identify possible blockchain security vulnerabilities and anomalies. Moreover, limited research has been conducted to assess blockchain security vulnerabilities and anomalies through testing on real systems. Most related works that exist have failed to recognise the importance of blockchain security vulnerabilities in actual blockchain systems.

In this paper, we seek to address this failure. We summarise the respective approaches and limitations in Table 1.

Our approach is based on experiments using real systems that can be used as a reference to secure similar blockchain-based sensor networks. Although researchers have identified blockchain as a potential security solution for IoT sensor networks, existing commercial blockchain applications are vulnerable to security threats, including physical threats. We consider this a critical concern. Our study shows that blockchain applications are still vulnerable to cyber security issues and low-power IoT devices may be susceptible to respective security concerns. Therefore, it is necessary to understand blockchain-related security vulnerabilities but also to identify associated anomalies that we can use to identify such attacks. We discuss the research methodology and test bed in the next section.

## 5. Research Methodology and Test Bed Development

Our study described in this paper combines practical experimentation with quantitative evaluation, conducting all tests within a controlled laboratory setting on a real test system. The experimental setup was developed using twenty Raspberry Pi and Orange Pi units [29,30]. Orange Pi Zero devices offer comparable specifications to the Raspberry Pi 3B models and support the same operating systems [29,30]. An operating system is typically required to install blockchain applications, resulting in minimum hardware capabilities to deploy. Many blockchain applications are 64-bit, and only a limited number of 32-bit applications are commercially available [13]. Widely deployed ARM Linux versions are typically 32-bit operating systems, and a minimum of 512 MB RAM and 4 GB space is typically required to install many 32-bit blockchain applications on ARM Linux distributions [30]. Not all low-power devices support the minimum hardware requirements to install an operating system and run blockchain platforms.

We chose to deploy Raspberry Pi and Orange Pi Zero devices because, unlike common microcontrollers, they support full computing capabilities, are capable of supporting general-purpose operating systems, have memory expansion capability, and support the processing power to run blockchain applications while remaining low-power and cost-effective [29].

We also tested Arduino Yun Rev 02 devices running the OpenWRT Linux distribution but found they had inadequate processing capacity to install and process blockchain applications. This is primarily due to the limited capability of the 16 MB Atheros AR9331 Linux microprocessor [29].

For the purposes of our blockchain network, we selected Hydrachain, Monero, and Duino Coin as our blockchain applications, installing them on each of the single-board computers [31]. These blockchain platforms were chosen because of their widespread use, compatibility with our hardware, efficient power usage, open-source software, and ease of programmability [31].

We also experimented with the Multichain, IPFS, Bitcoin, and Ethereum blockchain platforms. However, we ultimately opted not to use these platforms [10]. We found that the Command Line (CMD) version of the Multichain application was incompatible with the Linux version, and the Graphical User Interface (GUI) was only available with 64-bit architecture, which consumes more power compared to other blockchain applications. Furthermore, Ethereum and Bitcoin applications consume a higher amount of energy than Hydrachain, Monero, and Duino Coin blockchains [13]. Additionally, the Bitcoin platform does not offer private blockchain features for our test network data transmission, which is only available for financial purposes. Finally, IPFS was not suitable for our Local Area IoT test network as IPFS is more suitable for cloud data transactions [10].

Additionally, we incorporated low-energy sensors to gather and convey sensor information across the blockchain network, which we connected through a wireless router. We used a temperature sensor LM-35, distance sensor HC-SR04, rain sensor, thin film pressure sensor, and tri-axis digital tilt sensor as our primary sensors to gather sensor data. The sensors are compatible with Raspberry Pi and Orange Pi devices that are connected to blockchain nodes via jumper wires over General-Purpose Input/Output (GPIO) pins [30]. All blockchain applications are installed on the Linux operating system. We used SSH Putty software (64-bit-0.81 version) to access each blockchain node and implement the necessary configurations [31]. Figure 1 shows the blockchain sensor network.

Experimental data were collected while processing blockchain applications, and quantitative data analysis tools were used to analyse the collected data. We carried out a Distributed Denial of Service (DDoS) attack and noted that it caused a potential security vulnerability leading to hardware failures and associated anomalous behaviours [32].

We used the Nmap vulnerability assessment tool to identify potential network vulnerabilities while performing a DDoS attack [32]. Nmap is an open-source tool used for network security audits, risk analysis, and network scanning. Nmap detects unsecured open ports, applications, and their exploits [32]. Blockchains typically use Transmission Control Protocol (TCP) to transmit blocks, which can be a target of a DDoS attack. We also analysed Block Transaction Rate (BTR) anomalies, block loss anomalies, network bandwidth usage anomalies, network latency, and device temperature anomalies as the visible anomaly behaviours in a DoS attack and a damaged sensor [32]. We used blockchain data transaction logs to collect BTR and block loss data. We further used Linux MPSTAT, DSTAT, and Wireshark tools to monitor network bandwidth usage [33]. Additionally, we created python-based code to collect temperature data from IoT devices running blockchain applications and integrated it with the Linux “vcgencmd” tool [34]. In the next paragraphs, we discuss key hardware and software used to develop our test bed system.

### 5.1. Resources

The test bed consists of a number of different software and hardware resources. This section presents the resources used to implement our blockchain-based sensor network prototype.

#### 5.1.1. Raspberry Pi

The Raspberry Pi is an ARM-based, low-powered, single-board computer series used to develop prototypes. The Raspberry Pi series consists of various models, from Raspberry Pi model 05 to Zero [29]. All of these models can run the Linux-based Raspbian operating system. Each model has different RAM and processing capacities, such as 512 MB or 1 GB [29]. These devices have 40-pin headers for connecting sensors and a wireless LAN interface for networking. The Raspberry Pi devices are powered by an ARM cortex CPU [29]. The main advantages of using Raspberry Pi devices are as follows:Higher processing power;ARM Linux operating system support;Python programming compatibility;Wireless and Ethernet support.

#### 5.1.2. Orange Pi

Orange Pi devices are single-board computers powered by the Allwinner H616, a 64-bit quad-core Cortex A53 processor, and are engineered to deliver efficient computational performance across applications [30]. They are equipped with either 512 MB or 1 GB of DDR3 RAM, accommodating a range of computational needs [30]. In terms of connectivity, Orange Pi devices offer both Wi-Fi and Ethernet networking options [30]. The integration of a USB Type-C 5V interface, along with USB 2.0 ports, underscores the devices’ adaptability in interfacing with external peripherals.

Orange Pi devices exhibit broad compatibility with available ARM-based Linux operating systems such as Ubuntu, Debian, and Android, thereby serving for software development and experimentation [30]. The product range extends from the Orange Pi Zero to the Orange Pi 5B. Also, selected models further enhance connectivity through Bluetooth 5.0 and offer a 39-pin header for interfacing with General-Purpose Input/Output (GPIO), Universal Asynchronous Receiver Transmitter (UART), and Serial Peripheral Interface (SPI) modules [30].

#### 5.1.3. Hydrachain Blockchain

Hydrachain blockchain is specifically engineered to facilitate the creation and deployment of private blockchain networks [35]. It is an extension of the Ethereum platform, inheriting its robust smart contract capabilities while introducing features tailored for permissioned and consortium blockchain environments [35]. At its core, Hydrachain emphasises transactional throughput, scalability, and privacy, offering a deterministic consensus mechanism that is optimised for high-speed transaction processing in private networks [35]. The platform supports the creation of multiple chains, or “hydras”, each capable of operating with its own consensus rules and participant groups, thereby offering flexibility and scalability for various application needs [35].

#### 5.1.4. Monero Blockchain

The Monero blockchain technology is a decentralised application that uses a public distributed ledger system [36]. The Monero platform allows for adaptive block size growth to accommodate transaction volume, ensuring scalability and efficiency. This technology also provides anonymity and fungibility for data transactions, meaning that users can conduct transactions without revealing their identity or the transaction history to others [36]. Monero uses an open-source block protocol called Cryptonote, which allows for zero-knowledge proofs and ring signatures, ensuring the privacy and security of transactions [36]. Monero employs Ring Confidential Transactions (RingCT), which obscure the amount of Monero being transacted, adding another layer of privacy to the system. The platform uses a group of stealth public IP addresses to hide the legitimate private IP address, and users can optionally share encryption keys for auditing purposes. All transactions on Monero are validated through a network called RandomX [36].

#### 5.1.5. Duino Coin Blockchain

The Duino Coin blockchain is a relatively new platform promoting more inclusive participation across devices ranging from low-power microcontrollers and single-board computers like Raspberry Pi to older PCs and smartphones [37]. This blockchain uses Duino Coin Unique Consensus Operation–S1 (DUCO-S1), which dynamically adjusts the performance in accordance with the computational capability of each participating device and uses XXHASH algorithms to run blockchain on low-power devices [37]. The Duino Coin application uses the “Kolka System” to support underpowered transactions without causing any unnecessary difficulties [37]. The Duino Coin blockchain is also an open-source platform that is both cost-effective and energy-efficient. Data transactions are encrypted using SHA1 encryption. Although Duino Coin is a centralised blockchain, it provides decentralised options for users [37].

### 5.2. Data Collection Metrics

This test bed also collects a number of different system and performance metrics. This section summarises the data collection metrics that were deployed.

#### 5.2.1. Distributed Denial of Service Attack Anomalies

Distributed Denial of Service (DDoS) attacks are a common threat to blockchain networks. One significant metric for evaluating DDoS vulnerability is the possibility of such attacks targeting test bed prototype networks [38]. We conducted a vulnerability assessment using the Nmap tool to identify DDoS vulnerability, which allowed us to determine that the test bed is susceptible to the “Broadcast Avahi” DDoS attack [38]. We deployed a series of DDoS attacks 100 times using the “Ettercap” network security tool and measured the successful service denials of each blockchain test network. Ettercap is an open-source network security software that is typically used for DoS and DDoS attacks, eavesdropping, password capturing, Man-in-the-middle attacks, and spoofing. Cyber attackers frequently employ these attacks to compromise blockchain networks [38,39]. While Ettercap deployed DDoS attacks, we used Nmap software (Version 7.95-2) to scan open network ports, including blockchain TCP ports 8333, 9333, 9999, 22,556, and 30,303. The Nmap tool conducts a vulnerability assessment and generates a report showing which ports are open, the IP addresses of vulnerable nodes, and MAC addresses [39]. Under the DDoS attack anomalies, we evaluate the IoT hardware temperature variability that processes blockchain applications, blockchain network latency variability, and network bandwidth usage variability [40].

#### 5.2.2. Hardware Failure Anomalies

As a result of DDoS attacks, hardware failures were observed. Blockchain network end device hardware failures provide consequent early identification and enable actions to be taken for prevention of possible intrusions [41]. We evaluate Block Transaction Rate (BTR) and network block loss percentage metrics with incidents of end device hardware failures [41].

## 6. Results and Evaluation

In this section, we evaluate the potential anomaly behaviours of Hydrachain, Monero, and Duino Coin blockchain networks in an incident of Denial-of-Service attack and a physically damaged sensor node. We discuss the Denial-of-Service attack anomalies in the next section.

### 6.1. Network Latency Anomalies

Network latency is an important aspect of network performance that can be affected by a DDoS attack [42]. An increase in network latency can limit blockchain services and data transaction availability. Low-power IoT sensor networks are particularly vulnerable to Denial-of-Service attacks, and delayed data transactions can potentially damage a business’ operation [42]. Figure 2 illustrates blockchain network latency under typical circumstances and network latency variability of three blockchain networks during a DDoS attack.

The results indicate that for all blockchains, the network latency significantly in-creased, typically doubling during a DDoS attack. The network latency of the Hydrachain blockchain network increased from 400 ms to 800 ms, while Monero’s network latency increased from 550 ms to 1000 ms. The latency of the Duino Coin blockchain network increased by a larger factor from 400 ms to 1100 ms. This indicates that DDoS attacks involve flooding blockchain networks, creating network congestion, and causing increased network latency. Blockchains typically require higher processing power to maintain the blockchain network without failure, whereas IoT devices are typically limited in this regard. The DDoS attacks can cause resource exhaustion on targeted blockchain nodes, this can result in the targeted nodes to use a higher batch time-out to release the next chain of blocks and ultimately increase the block time.

Our results demonstrate that Distributed Denial of Service attacks can critically affect the latency of blockchain networks, leading to a potential shutdown of all blockchain data transactions while increasing network bandwidth usage.

### 6.2. Network Bandwidth Usage Anomalies

The bandwidth of an IoT wireless network is another critical metric that can be targeted by a DDoS attack. When a DDoS attacker generates excessive network traffic, it can impact bandwidth usage [43]. The anomalous behaviour of bandwidth usage is a key indicator that users can use to identify a DDoS attack threat. Higher bandwidth usage can also impact the blockchain services of blockchain networks [43].

Figure 3 presents the network bandwidth usage under typical circumstances and victim blockchain networks during a Denial-of-Service (DoS) attack. The results indicate that the bandwidth usage significantly increased compared to the bandwidth usage of the three blockchain networks under normal circumstances [44].

The Hydrachain blockchain network typically uses bandwidth ranging from 1400 to 1600 Kbps. However, bandwidth usage increased, ranging from 2000 to 2200 Kbps during a DDoS attack. The Monero blockchain network’s usual bandwidth usage is between 80 and 160 Kbps. However, during a DDoS attack, the victim Monero blockchain network’s bandwidth usage was recorded as between 1700 and 1900 Kbps.

The bandwidth usage of a Duino Coin blockchain network varies from 50 to 300 Kbps under normal circumstances. However, during a DDoS attack, the bandwidth usage increased to 1800–2000 Kbps. The victim Hydrachain network had a mean bandwidth usage of 2121.5 Kbps, while the Monero and Duino Coin victim networks had a mean bandwidth usage of 1809.5 Kbps and 1900 Kbps, respectively.

Due to both increased network congestion and resource exhaustion, DDoS attacks can result in higher bandwidth utilisation of the blockchain network. Our blockchain nodes were connected over Wi-Fi, which we would expect many IoT devices to use for network connectivity. The increase in bandwidth consumption during a DDoS attack can impact other aspects of the blockchain node. Increased power usage during an attack can be expected from the increased compute load and network utilisation. We expect that this change in load may cause an increase in the device temperature.

### 6.3. Device Temperature Anomalies

It is not widely appreciated how susceptible IoT hardware devices are to failure caused by high operating temperatures [45]. We observed that during a DDoS attack, the temperature of the victim node increased dramatically, which could potentially cause hardware failure due to overheating [45]. This can also pose a risk of fire hazards. It is crucial to identify any temperature variability anomalies that may occur during a DDoS threat [45]. We individually analysed how device temperature varies based on three blockchain platforms.

Typically, blockchain platforms use CPU and GPU cores to process data blocks. Due to the low processing power of IoT devices, any increase in the demands on the system may result in a subsequent increase in hardware temperature [40]. A number of events may result in increased load. One such cause is constant network congestion due to a DDoS attack, which causes the blockchain to generate an increased number of hashes and to re-lease them to the network, resulting in higher CPU and GPU utilisation [40]. A second cause relates to the fact that the blockchain ledger keeps every sensor data transaction record in each hardware node locally and, during a DDoS attack, the ledger requires increased processing power and time to update the victim node records [40].

Figure 4 displays the temperature variability of each node in the Hydrachain blockchain network devices that also includes a victim node (node 10). Under typical circumstances, the temperature range of the Hydrachain blockchain network device is between 66 °C and 68 °C. However, the temperature of the victim node is significantly higher, with the device temperature ranging between 78 °C and 80 °C. Also, the mean temperature of non-victim nodes ranges from 65.42 °C to 67.21 °C, while the victim node has a mean temperature of 79.59 °C. This indicates a significant difference in temperature between the victim node and the other blockchain nodes.

Figure 5 shows the temperature variability of each node in the Monero blockchain network that includes a victim node (node 10). We can see that the temperature of the device connected to the victim node in the Monero blockchain network reached 72 °C. Meanwhile, the non-victim nodes show device temperature variabilities between 58 °C and 60 °C. Also, the mean temperature of non-victim nodes ranges from 57.63 °C to 59.21 °C, while the victim node has a mean temperature of 71.02 °C.

From Figure 6, we see that the temperature of the Duino Coin victim node exceeded 90 °C, while the non-victim nodes showed device temperature variability between 82 °C and 84 °C. Also, the mean device temperatures for non-victim nodes range from 82.54 °C to 83.13 °C, while the victim node has a mean temperature of 90.94 °C.

Considering the results for all blockchains, we note a varying range of average operating temperatures; however, there is a significant increase in operating temperature for the node under attack. We note that the recommended operating conditions for the respective devices are between 0 °C and 85 °C [29]. However, the recommended operating device temperature for a Local Area Network (LAN) is from 0 °C to 70 °C [29]. As highlighted by the manufacturer, there is a possibility of hardware failure if the device temperature exceeds these bounds.

IoT devices are typically designed to operate in an ambient room temperature of 20–30 °C. If these devices constantly run at higher temperatures or with maximum CPU and GPU levels, the single-board devices may shutdown to prevent any hardware component damages [29]. Regardless, constant higher temperatures can result in hardware failure and the manufacturer has recommended thermal management systems such as a heatsink, a fan, or liquid cooling if devices are run 24/7 [29].

While all normal nodes maintained a temperature under 70 °C, all victim nodes recorded average temperature levels above the recommended maximum temperature of 70 °C. The average temperature reached as high as 90.94 °C for the Duino Coin victim node, well above the recommended maximum temperature for an individual device.

The temperature increase seen for all three blockchains has the potential to cause a hardware device failure. This indicates that low-power IoT hardware device functionalities can potentially be impacted by excessively high temperature values.

In summary, Distributed Denial of Service attacks increase network latency and network bandwidth, leading to higher temperature values of low-power IoT devices and potentially causing hardware failures and damages.

### 6.4. Block Transaction Rate Anomalies

As previously noted, the increased temperature caused at a node during a DDoS attack has the potential to lead to hardware failure. Next, we would like to explore the impact of a hardware failure on the blockchain network.

The Block Transaction Rate (BTR) changes significantly when a sensor node is damaged. In this section, we present the anomalous BTR behaviour of a damaged sensor node, along with four other blockchain nodes, for analysis purposes [46]. We have received similar results from the analysis of blockchain networks with twenty blockchain nodes. As part of our testing process, we emulated a node sensor failure by intentionally damaging the connector on the sensor node breadboard.

It is important to note that physical damage to sensor nodes can also occur due to factors such as high temperatures, electrical shorts, human error, and water damage [47]. These types of damages should be considered when deploying any network in a real environment [47]. We used a damaged sensor node to analyse the anomaly behaviour of all three blockchain networks. The test was conducted for 100 h.

Figure 7 plots the BTR for a five-node Hydrachain blockchain network. Undamaged nodes display a BTR ranging from 10 to 20 blocks per second. However, the damaged blockchain node gradually decreases its BTR within the first 30 s, ultimately leading to a complete halt of all transactions.

Figure 8 plots the same results for a Monero blockchain network. The normal BTR is between 20 and 50 blocks per second. However, in this network as well, the malfunctioning blockchain node reduces transactions within the first 20 s and eventually stops transmitting blocks.

Finally, Figure 9 plots the results for the Duino Coin blockchain. The normal BTR ranges from 15 blocks per second to 25 blocks per second. The results indicate that a damaged sensor node can have a significant impact on the block transaction rate [47]. The transaction rate starts to decrease within the first 30 s and eventually stops completely.

All these results highlight the potential consequences of a physically damaged sensor node on the blockchain network, regardless of whether the node fails due to natural causes or DDoS attack [47].

### 6.5. Block Loss Anomalies

In an IoT blockchain network, losing sensitive data due to block loss can be a serious issue in an environment like a hospital facility, which may also lead to life risks [48]. A physically damaged sensor node can increase block loss. To prevent such incidents, blockchain users should analyse block loss anomalies and undertake suitable monitoring [48,49]. The block loss anomalies of a physically damaged sensor node are illustrated in Figure 10, Figure 11 and Figure 12 for each of the three blockchains being evaluated. We used a sample of 1000 blocks to analyse the behaviour of block loss anomalies. We found that each blockchain network experiences significant block loss when there is a physically damaged sensor node.

In the Hydrachain blockchain network shown in Figure 10, 325 blocks out of the 1000 blocks sampled were successfully transmitted, which accounts for 32.5 percent of the total block transactions. Similarly, in the Monero blockchain network (Figure 11), the damaged sensor node could only transmit 187 blocks successfully out of the 1000-block sample, which is only 18.7 percent of the total block transaction.

The damaged sensor node in the Duino Coin blockchain network was able to transmit significantly more blocks with 351 successful transfers, but still low at 35.1 percent of the total block transaction, as illustrated in Figure 12.

Although all networks with damaged sensor nodes lost blocks, the Duino Coin blockchain network had a greater success rate [49]. However, the results indicate that damaged hardware is a prime factor for losing data blocks, regardless of the blockchain architecture.

## 7. Conclusions and Future Research

The use of blockchain technology in IoT-based low-power sensor networks can address many network security issues and concerns [50]. However, both blockchain technology and IoT are emerging research areas, and there are several areas that require attention. Although blockchains offer promising security features such as cryptography, ledger function, and decentralisation, blockchain-based IoT low-power sensor networks can still face potential security threats from DDoS attacks [50]. In this paper, we analysed the potential security vulnerabilities that blockchain-based IoT sensor networks may face as a result of a DDoS attack and some of the consequent anomalies. We conducted this analysis using a test bed made up of widely used IoT hardware that allowed us to understand the real environment challenges [50].

To collect data, we conducted a vulnerability assessment of Hydrachain, Monero, and Duino Coin blockchain networks using the Nmap software tool while performing a DDoS attack using the Ettercap tool [51,52]. During the assessment, we identified that blockchain networks are still vulnerable to Distributed Denial of Service attacks and that potential hardware failures can occur due to DDoS attacks. The results show that DDoS attacks cause an unusual temperature increase in IoT end devices, which causes hardware failures while increasing blockchain network latency and bandwidth usage compared to non-victim nodes [52].

We observed that all three blockchain networks exhibit noticeable anomalies that can be used to detect potential network security threats [53]. The anomalies were analysed with respect to both DDoS attacks and physical device damages. Our results indicate that blockchain network latency can be impacted by DDoS attacks, causing significant delays such as 800 ms and 1100 ms. Also, the experiments show us that DDoS attacks cause a significant rise in the temperature of hardware devices that process blockchain networks, which can exceed 90 °C and potentially lead to device failures, and can thus cause fire hazards and loss of sensitive data blocks [53]. The network also shows considerable variability in bandwidth usage that can increase to 2200 Kbps and drop the Block Transaction Rate to zero blocks/sec compared to non-victim nodes. The hardware failures can also lose as much as 81.3 percent of data blocks, which is a critical consideration.

The study can be extended to evaluate other possible blockchain attack vulnerabilities, such as Sybil attacks, routing attacks, dictionary attacks, and time jacking. These attacks can cause security breaches in low-power IoT networks [53]. Our work clearly identifies that when a single node is a victim of a DDoS attack, the temperature anomaly compared to other nodes in the blockchain is significant. This should be used as an avenue to expand on in further research to determine whether these anomalies are also significant as a larger proportion of blockchain nodes become DDoS victim nodes. This may help to identify possible avenues to detect DDoS attacks in this environment. The use of blockchain technology on low-power IoT devices could lead to new research opportunities in the future. One possible avenue is to explore low-power hardware technologies [54]. Since most IoT devices are low powered, it could be a new research area to investigate how to use blockchain with them. Another possibility is to develop blockchain platforms that target different low-power microcontroller devices that consume low hardware resources and energy [54].

Additionally, future developments could incorporate various blockchain applications and wireless technologies such as 5G, 6G cellular networks, Bluetooth, ZigBee, and LoRa [55]. This could improve the use of blockchain technology to secure information in various industries like healthcare, supply chain, automotive, and agriculture [55].

Finally, analysing security vulnerabilities and anomalies is significant for deploying blockchain-based sensor networks; yet, many research problems must be explored while addressing the current research issues.

## Figures and Tables

**Figure 1 sensors-24-03083-f001:**
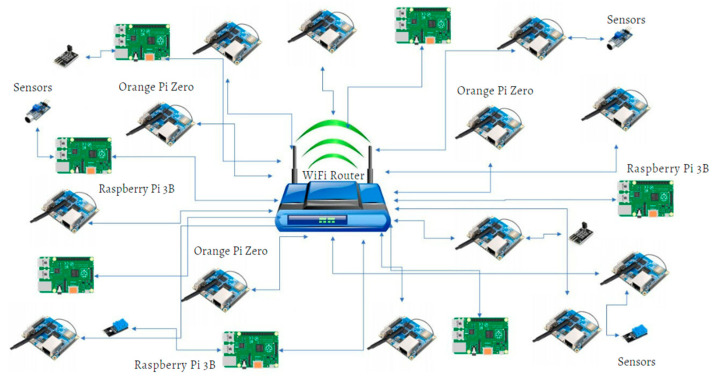
Sensor network architecture.

**Figure 2 sensors-24-03083-f002:**
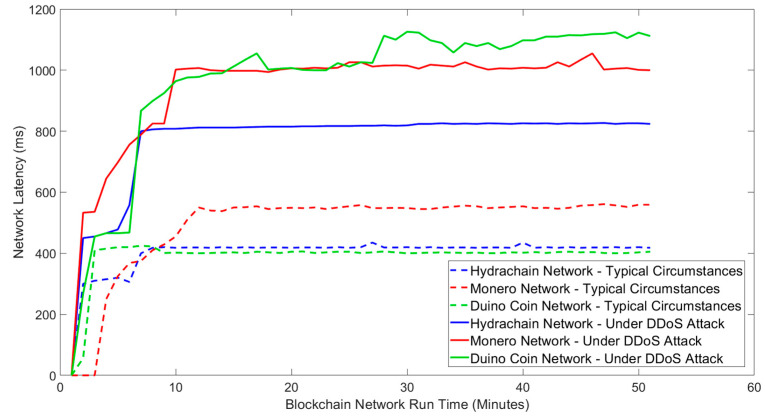
Latency variability of blockchain networks under typical circumstances and DDoS attack.

**Figure 3 sensors-24-03083-f003:**
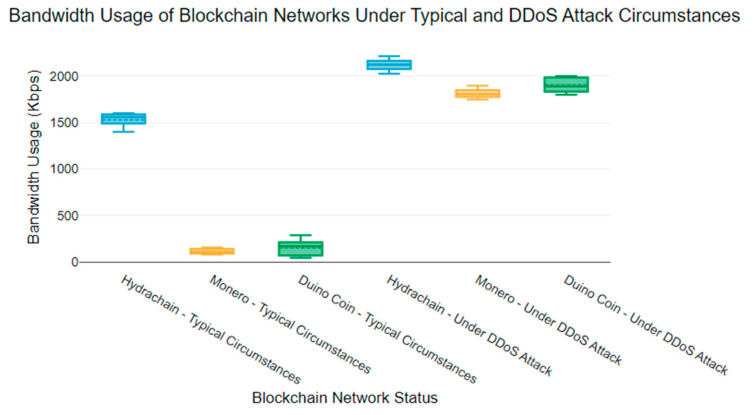
Bandwidth usage of blockchain networks under typical circumstances and a DDoS attack.

**Figure 4 sensors-24-03083-f004:**
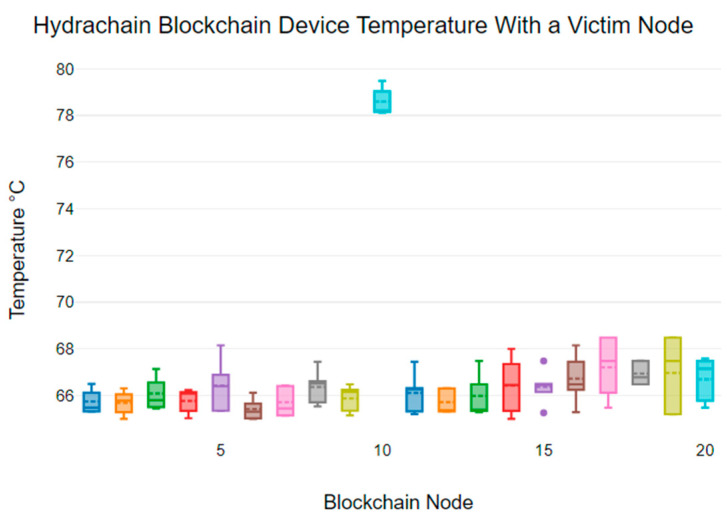
Node temperatures for Hydrachain: node 10 targeted for DDoS.

**Figure 5 sensors-24-03083-f005:**
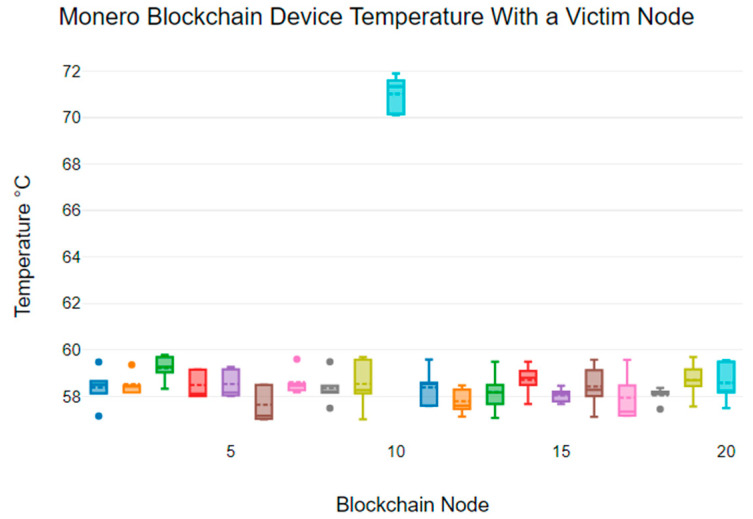
Node temperatures for Monero: node 10 targeted for DDoS.

**Figure 6 sensors-24-03083-f006:**
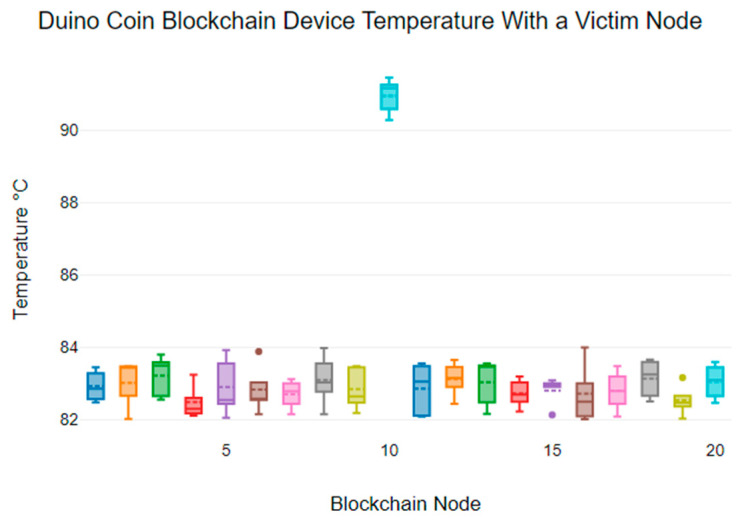
Node temperatures for Duino Coin: node 10 targeted for DDoS.

**Figure 7 sensors-24-03083-f007:**
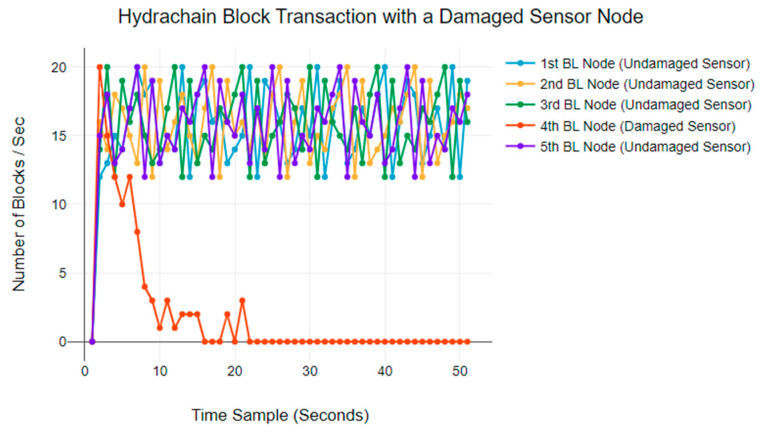
Hydrachain blockchain BTR anomaly behaviours.

**Figure 8 sensors-24-03083-f008:**
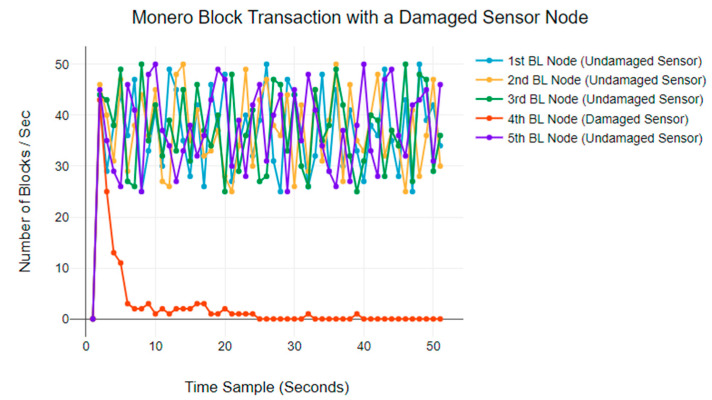
Monero blockchain BTR anomaly behaviours.

**Figure 9 sensors-24-03083-f009:**
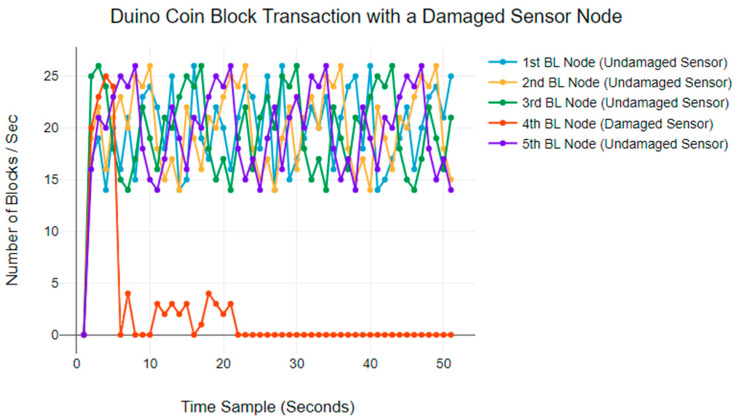
Duino Coin blockchain BTR anomaly behaviours.

**Figure 10 sensors-24-03083-f010:**
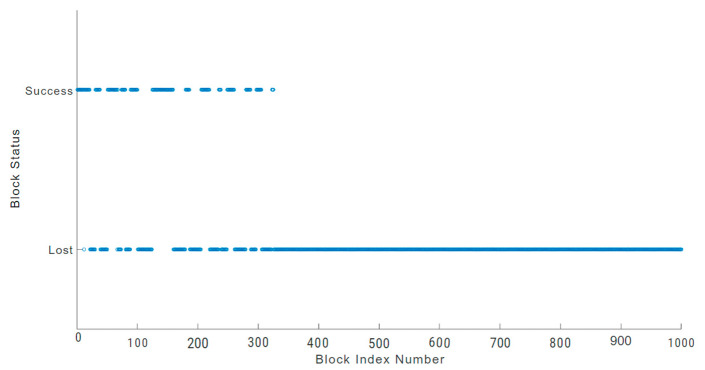
Hydrachain blockchain block loss.

**Figure 11 sensors-24-03083-f011:**
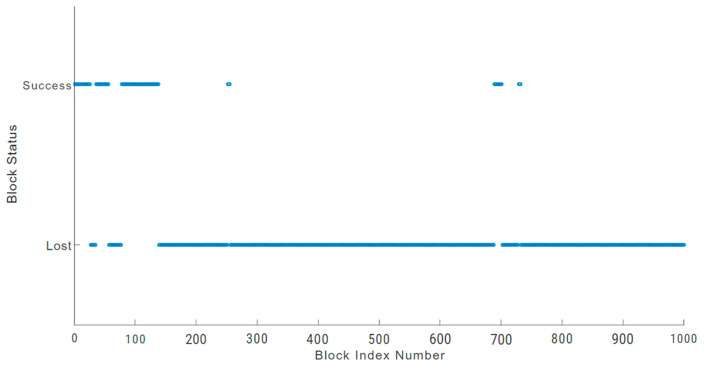
Monero blockchain block loss.

**Figure 12 sensors-24-03083-f012:**
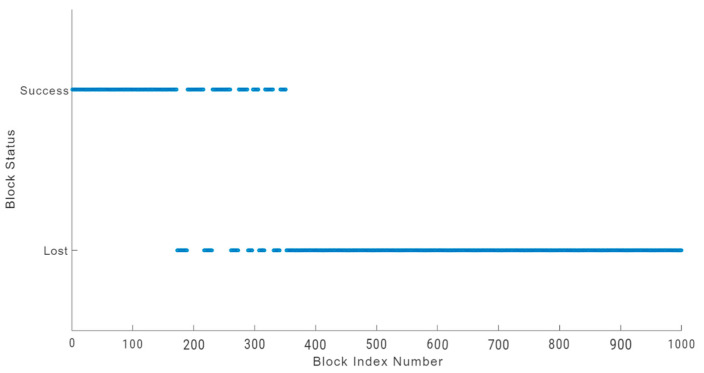
Duino Coin blockchain block loss.

**Table 1 sensors-24-03083-t001:** Approaches and limitations in existing research work.

Approach	Limitations
Propose a new theoretical model to integrate blockchain and IoT technologies called BCoT [14]	The theoretical model ignores the practical challenges for integrating blockchains and low-powered IoT devices.
Proposed a new blockchain trust model for wireless sensor networks [17]	The proposed trust model is untested in a real blockchain-based wireless test system to identify performance limitations and vulnerabilities.
Conducted a simulation to exploit the Bitcoin and Ethereum blockchain mining pools to understand the vulnerabilities [19]	The simulation only focused on cryptocurrency mining pools other than the real IoT hardware performance and network vulnerabilities.
Proposed the Hyperledger Fabric platform to detect IoT network anomalies [20]	Although Hyperledger Fabric is a solution to detect anomalies, the research has ignored the IoT hardware limitations and processing power requirements.
Proposed a new model called the “integrity protection service model” to secure digital health systems [21]	The proposed work only considered the theoretical capabilities of blockchains; practical integration challenges are ignored. In particular, security challenges in real systems are missed.
Proposed Hyperledger Fabric as an operating system to monitor the integrity level of data transmission and secure accessibility [24]	The proposed solution is more suitable for devices with higher processing power as the proposed solution ignores the power consumption and temperature metrics of low-powered devices.
Suggested blockchains as a robust security solution to mitigate security threats [22,23,28]	Although blockchains are suggested as a robust security solution, security threats on blockchain networks are ignored.
Proposed a blockchain-based framework called “ELSO” for Health Information Exchange systems [26]	The proposed framework has missed possible security challenges in low-powered devices and the processing power requirement to run GUI blockchain platforms in real IoT devices.
Proposed a blockchain-based deep learning model called “DeepChain to protect Health Information Exchange [27]	The model has only been simulated in a virtual machine environment and the proposed solution may generate comparatively different results to a real IoT low-powered network.

## Data Availability

All datasets generated during the study are available upon request from the primary author.

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
