# Peer review of "An Analysis of Blockchain-Based IoT Sensor Network Distributed Denial of Service Attacks"

_sensors, 2024, doi:10.3390/s24103083_

Round 1

Reviewer 1 Report

Comments and Suggestions for Authors

1. What are the operating temperature and maximum temperature the mentioned devices can handle? It would be great to compare their datasheet-based operating parameters with the reported temperature.

2.  Typically, DDoS attacks are handled at the network level, and underlying IoT devices don't often receive so many packets that might lead to a temperature rise. The setting is unrealistic in real-world scenarios. 

3. The motivation is not clear. A use case or real-world scenario would be beneficial for the work, especially as DDoS mitigation is a well-established research field and many different solutions exist.

Comments on the Quality of English Language

N/A

Author Response

Dear Reviewer,

Reviewer 2 Report

Comments and Suggestions for Authors

The authors used a testbed to demonstrate that blockchain integrated in IoT platforms are vulnerable to DDoS attacks. Overall, the paper is well organized and easy to understand. The obtained results look important. However, I have the following comments regarding the proposed work:

1) Please add a table at the end of the related work section to summarize the state of the art techniques while showing their limitations. Additionally, some important and recent references are missed. I suggest to add the below references:

ELSO: A Blockchain-Based Technique for a Reliable and Secure Healthcare Information Exchange

DeepChain: A Deep Learning and Blockchain Based Framework for Detecting Risky Transactions on HIE System

Octa Pillars-based Approach to Select the Best Blockchain-based Solutions in Healthcare Information Exchange

2) Please justify the importance of selecting heterogeneous devices in Figure 1. 

3) Provide more information about the experiment setup.

4) In most figures, the authors explain what they observe on the figures without giving any explanation why those results are obtained.

5) The authors limited their experiment in figures 4, 5 and 6 to detecting only 1 victim one. What happen in case several nodes are considered as victims at the same time?

6) I miss the research aspect in the paper where the authors more focused on the application aspect.

7) A major weakness of the paper is that the authors are not sufficiently compared their wok to the state of the art techniques.  

Comments on the Quality of English Language

 Minor editing of English language required

Author Response

Dear Reviewer,

Round 2

Reviewer 1 Report

Comments and Suggestions for Authors

The manuscript has improved significantly.

Reviewer 2 Report

Comments and Suggestions for Authors

The authors answered all my concerns and comments. The quality of the paper is highly improved. I recommend  the paper for publication.